# The Effects of Grapevine (*Vitis vinifera* L.) Leaf Extract on Growth Performance, Antioxidant Status, and Immunity of Zebrafish (*Danio rerio*)

Seyed Hossein Hoseinifar [1,*,†], Zohreh Fazelan [2,†], Ehab El-Haroun [3], Morteza Yousefi [4], Metin Yazici [5], Hien Van Doan [6] and Marina Paolucci [7]

1 Department of Fisheries, Faculty of Fisheries and Environmental Sciences, Gorgan University of Agricultural Sciences and Natural Resources, Gorgan 49189-43464, Iran

2 Department of Animal Science and Aquaculture, Faculty of Agriculture, Dalhousie University, Truro, NS B2N 5E3, Canada; zohreh.fazelan@dal.ca

3 Fish Nutrition Research Laboratory, Department of Animal Production, Faculty of Agriculture, Cairo University, Cairo 12411, Egypt; ehab.elharoun@kysu.edu

4 Department of Veterinary Medicine, Peoples' Friendship University of Russia (RUDN University), 6 Miklukho-Maklaya St., Moscow 117198, Russia; yousefi-m@rudn.ru

5 Faculty of Marine Sciences and Technology, Iskenderun Technical University, Iskenderun 31200, Türkiye; metin.yazici@iste.edu.tr

6 Department of Animal and Aquatic Sciences, Faculty of Agriculture, Chiang Mai University, Chiang Mai 50200, Thailand; hien.d@cmu.ac.th

7 Department of Sciences and Technologies, University of Sannio, 82100 Benevento, Italy; paolucci@unisannio.it

* Correspondence: hoseinifar@gau.ac.ir

† These authors contributed equally to this work.

**Abstract:** An 8-week feeding trial was carried out to evaluate the effects of grapevine (*Vitis vinifera*) leaf extract (GLE) on the growth, oxidative enzymatic activities, immunity, and expression of antioxidant genes in zebrafish (*Danio rerio*). Three hundred and sixty zebrafish were supplied and fed with different levels of GLE: 0, 0.5, 1, and 2 g kg$^{-1}$. The dietary administration of 1 g kg$^{-1}$ of GLE significantly increased growth parameters in fish. Fish fed diets with GLE showed increased total protein. The total Ig and lysozyme activity significantly changed in the whole-body serum, but not in skin mucus. GLE significantly increased Catalase (CAT), Superoxide Dismutase (SOD), and Glutathione Peroxidase (GPx) activities compared to the control diet. GLE treatments caused a significant decrease in the malondialdehyde (MDA) content. In the skin mucus, only CAT and SOD activities significantly increased. The highest expression of Toll-like receptor-1 (TLR-1) and Tumor Necrosis Factor-$\alpha$ (TNF$\alpha$) genes was achieved in fish fed 2 g kg$^{-1}$ of GLE. CAT and SOD gene expressions were significantly higher in fish fed 1 and 2 g kg$^{-1}$ of GLE. GPx gene expression was significantly higher in fish fed 1 g kg$^{-1}$ of GLE. In conclusion, the results of the present study revealed that GLE affects growth performance and regulates antioxidant and immune gene expression. The determination of the optimum dosage merits further research.

**Keywords:** antioxidant genes; growth; immunity; immunity genes; polyphenols; *Vitis vinifera*; zebrafish

**Key Contribution:** Grapevine leaf extract has the potential to be used as a feed additive in zebrafish to boost growth performance and regulate antioxidant and immune gene expression.

## 1. Introduction

Polyphenols are a group of phytochemicals, secondary metabolites, present in fruits and vegetables and synthesized to control pathogen infections and to dissuade herbivorous animals from eating them [1]. In recent years, polyphenols have stirred interest due to their antioxidant properties, which enable them to complement the functions of vitamins and enzymes in the defense against oxidative stress caused by reactive oxygen species or

free radicals, generated during the cell metabolic processes [2]. Food and nutraceutical industries have largely exploited the antioxidant power of polyphenols and their protective effects on the cardiovascular and nervous system and against diseases such as cancer and diabetes, and pathological conditions such as obesity and inflammation [3,4]. The multifaceted polyphenol effects have been also exploited in terrestrial animal farming, due to their antimicrobial, growth-promoting, anti-inflammatory, and antioxidant properties, while acting as natural antibiotics [5–7]. However, the effects of polyphenols are not generalizable. Their chemical structure is highly variable and, with it, the type of activity. Polyphenols are highly diverse compounds that include more than 8000 different chemical types divided into several subtypes, ranging from fairly basic substances (such as phenolic acids) to complex molecules (such as tannins). Based on their chemical structure (in particular, on the number of aromatic rings contained and the structural elements that bind the aromatic rings together), several classes that include flavonoids, phenolic acids, stilbenes, and lignans are identified [8].

Interestingly, some polyphenols show pro-oxidant properties and other harmful effects depending on a variety of factors such as structure, concentration, presence of metal ions, microenvironment inside different tissues, and duration of administration [9]. Just to name a few, catechins from grape seeds show in vitro prooxidant activity on leukocytes [10]. Curcumin administered to humans behaves as a pro-oxidant depending on the body's location, age, sex, genetic structure, and concentration [11]. Resveratrol shows cytotoxic and pro-oxidant effects on hepatic cells depending on its concentration and time of exposure [12]. Moreover, it shortens the lifespan of *Saccharomyces cerevisiae* by a pro-oxidant mechanism [13].

The application of polyphenols in aquaculture is still in its infancy; however, several investigations indicate that they can be considered as feed additives with useful applications as growth promoters, antioxidants, and immune boosters, as well as acting as natural antibiotics [14,15]. Indeed, polyphenols from the chestnut shell and olive mill wastewater (OMWW) improved weight gain and stimulated the immune response and the antioxidant properties in Nile tilapia (*Oreochromis niloticus*) [16], juvenile beluga sturgeon (*Huso huso*) [17], common carp (*Cyprinus carpio*) [18], and convict cichlid (*Amatitlania nigro-fasciata*) [19]. However, diets supplemented with 10, 20, or 30 g kg$^{-1}$ of tannic acid resulted in poor growth performance in juvenile European seabass (*Dicentrarchus labrax* L.) [20]. The administration of diets containing 10 and 50 g kg$^{-1}$ of OMWW caused growth inhibition of gilthead sea bream (*Sparus aurata*) [21] and rainbow trout (*Oncorhynchus mykiss*) [22,23]. Further, studies carried out on zebrafish show that the efficacy of polyphenols in counteracting intestinal inflammation depends not only on the doses employed [24] but also on the timing of the treatment [25].

The emerging picture is a highly variable one, indicating that it is not possible to generalize the type of polyphenols and the actions performed and that each polyphenol or mixture of polyphenols must be characterized and evaluated for the effect exerted. In this frame, the zebrafish—a widely employed biomedicine and, more recently, aquaculture model [26,27]—may represent an extraordinary resource to evaluate the efficacy of polyphenols in aquaculture, by limiting the costs of experimentation and providing basic but crucial data such as suitable concentrations and biological effects, which can be defined and later adapted to the species of commercial interest. Moreover, this approach would comply with the principles of the 3R (replacement, reduction, and refinement) in animal experimentation, proposed by [28] and at the basis of the Directive on the protection of animals used for scientific purposes of the EU (2010/63/EU, law on 22 September 2010).

The use of phytochemicals in aquaculture to promote growth and control disease outbreaks and microbe pathogen infection is a promising tool with minimum environmental impact. Thus, we undertook this study to evaluate the role of polyphenols extracted from *Vitis vinifera* L. leaf on the growth performance, antioxidant status, and immunity of zebrafish (*Danio rerio*).

## 2. Materials and Methods

### 2.1. Ethics

The experiments were performed in compliance with the protocols (357; 8 November 2000) approved by the ethics committee of the faculty of sciences of the University of Tehran.

### 2.2. Experimental Design and Feed Formulation

The proximate composition of the basal diet (control diet) is reported in Table 1. It was a commercial diet (Biomar, Nersac, France) used in previous studies with zebrafish. The basal diet was supplemented with *Vitis vinifera* leaf spray-dried extract (Grapevine leaf extract = GLE) supplied by a company specializing in the production of botanical extracts. According to the manufacturer's technical sheet (EPO Istituto Farmochimico Fitoterapico S.r.l., Milan, Italy), the extract was obtained by using warm water as the extraction solvent. The composition of the extract was carbohydrates 95–97%, lipids 1–2%, proteins 0–1%, minerals 1–2%, and polyphenols 5%. Three hundred and sixty healthy zebrafish were purchased from a private company (Gorgan Mahi, Gorgan, Iran). Before the commencement of the feeding trial, zebrafish were adapted for 14 days to the lab conditions and fed with the control diet three times per day. Zebrafish were weighed (see Section 3.1 "Growth performance" for initial weights) and divided into 12 aquaria of 100 L with a capacity of 30 fish/aquaria and treated as follows: (1) control group fed with the basal diet; (2) group fed with the basal diet added with 0.5 g kg$^{-1}$ of feed of *Vitis vinifera* leaf extract (GLE1); (3) group fed with the basal diet added with 1 g kg$^{-1}$ of feed of *Vitis vinifera* leaf extract (GLE2); (4) group fed with the basal diet added with 2 g kg$^{-1}$ of feed of *Vitis vinifera* leaf extract (GLE3). The preparation of the experimental diets was performed as described in our previous paper [29]. Briefly, the commercial diet was powdered and desired levels of GLP were added at the expense of cellulose (to keep the total energy of different diets identical). Then, they were re-pelleted and grounded to produce a suitable crumble (1 mm). All diets were maintained at 4 °C until use. The trial length was 8 weeks. During the feeding trial, fish were fed up to apparent satiation. Care was taken to minimize feed loss. The tank conditions were as follows: water DO (7.8 ± 0.2 mg L$^{-1}$), temperature 26.50 ± 1.20 °C, and pH 7.02 ± 0.3.

**Table 1.** Proximate composition of the basal diet.

| Proximate Composition (%) | |
| --- | --- |
| Dry matter | 93.6 |
| Crude protein | 38.9 |
| Crude lipid | 15.0 |
| Ash | 11 |

### 2.3. Growth Performance

Growth performance parameters were determined following the formula below. To determine the weight of fish, all fish were anesthetized with clove powder (250 mg/L) and weighed.

$$\text{Weight Gain (WG)} = \text{final body weight (FW)} - \text{initial body weight (IW)}$$

$$\text{Specific growth rate (SGR)} = 100 \times [(\ln \text{FW} - \ln \text{IW})/\text{the length of feeding trial}]$$

where IW is the initial weight and FW is the final weight.

### 2.4. Determination of Innate Immune and Antioxidant Parameters

Given the small size of the fish at the end of the trial, it was not possible to collect blood and serum. Therefore, at the end of the feeding trial, nine fish were sampled per

treatment, and a whole-body extract (WBE) was obtained as reported in [29,30]. Skin mucus (n = 9) was collected using a plastic zip pack as described in [31]. To determine whole-body serum lysozyme activity, we used a lysozyme-sensitive bacteria (*Micrococcus luteus* (PTTC)), which was determined following [32] by using a turbidimetric assay. The decrease in absorbance by each 0.001 min$^{-1}$ was considered a unit of lysozyme activity. Total protein was determined according to the standard method of Lowry [33]. The WB serum Ig levels were determined according to [34] using polyethylene glycol (Sigma-Aldrich, Saint Louis, MO, USA). Briefly, the method is based on precipitation decreasing immunoglobulin molecules using polyethylene glycol and the determination of total Ig by re-measuring total protein. The activity of antioxidant enzymes including Superoxide Dismutase (SOD), Catalase (CAT), and Glutathione Peroxidase (GPx) was determined by using commercial kits (ZellBio GmbH, Lonsee, Germany) according to the manufacturer's instructions. Malondialdehyde (MDA) was evaluated by the calorimetric method after [31].

### 2.5. Sampling for Gene Expression Study

At the end of the study, nine fish from each treatment (3 fish per tank) were randomly sampled and sacrificed with a clove solution (1000 mg L$^{-1}$). The intestine was dissected and instantly put in liquid nitrogen. Intestine samples were then transferred to a $-80$ °C refrigerator until further analysis.

### 2.6. RNA Isolation and Laboratory Methods of Gene Expression

The total RNA of samples was extracted and cDNA was synthesized as described in our previous paper [35]. Briefly, the isolation of total RNA was performed by using BIOZOL Reagent. Total RNA was treated with DNase I (Fermentas, Vilnius, Lithuania) to remove DNA. After checking and confirming the quality and quantity of isolated RNA with a Nanodrop Spectrophotometer and 1.5% Agarose gel, cDNA samples were prepared by using the DNA synthesis kit (Invitrogen, Waltham, MA, USA). To determine the expression levels of genes, primers were designed with reference to the Gene Bank sequences (Table 2) by Primer3 software. Then, the real-time PCR analysis and normalization of expression levels were performed as described in [36]. The glyceraldehyde-3-phosphate dehydrogenase (Gapdh) was considered a housekeeping gene.. The IQ5 optical system software (Bio-Rad, Hercules, CA, USA) and $\Delta\Delta$Ct method were used for data analysis.

**Table 2.** Sequences of the primers used to study the expression of selected immune and antioxidant-related genes expression in Zebrafish. TLR-1 = Toll-like receptor-1, TNF$\alpha$ = Tumor Necrosis Factor-$\alpha$, SOD = Superoxide Dismutase, CAT = Catalase, GPx = Glutathione Peroxidase, Gapdh = glyceraldehyde-3-phosphate dehydrogenase.

| Primer Name | Primer Sequence | Application | Accession Number |
|---|---|---|---|
| TLR-1 q-PCRF | CAGAGCGAATGGTGCCACTAT | immune | AY389444 |
| TLR-1 q-PCRR | GTGGCAGAGGCTCCAGAAGA | | |
| TNF-$\alpha$-PCRF | CTGCTTCACGCTCCATAAGA | immune | AY427649 |
| TNF-$\alpha$-PCRR | CTGGTCCTGGTCATCTCTCC | | |
| GPx q-PCRF | CCAAGTAAACCAGCGGCTTCT | antioxidant | NM_001007281 |
| GPx q-PCRR | TGATGTGCAGTGGACGGTTTAT | | |
| SOD q-PCRF | GGGTGGCAATGAGGAAAG | antioxidant | BC055516 |
| SOD q-PCRR | GCCCACATAGAAATGCACAG | | |
| CAT q-PCRF | GCATGTTGGAAAGACGACAC | antioxidant | AJ007505 |
| CAT q-PCRR | GTGGATGAAAGACGGAGACA | | |
| Gapdh q-PCRF | GTGGAGTCTACTGGTGTCTTC | Housekeeping gene | BC083506 |
| Gapdh q-PCRR | GTGCAGGAGGCATTGCTTACA | | |

*2.7. Data Analysis*

The study was designed as a complete randomized design including four treatments in triplicates. Data are presented as mean ± S.D. First, the normality of the data was checked and confirmed using the Kolmogorov–Smirnov test. To determine statistically significant differences at $p < 0.05$, data were subjected to one-way ANOVA followed by Duncan's multiple tests. The statistical analysis was performed using SPSS (version 16, SPSS, Armonk, NY, USA).

## 3. Results

*3.1. Growth Performance*

The results of GLE on growth performance are reported in Table 3. Zebrafish fed with 1 g kg$^{-1}$ of GLE showed higher values of FW, WG, and SGR compared with GLE2, GLE3, and control groups ($p < 0.05$). Zebrafish fed with 0.5 and 2 g kg$^{-1}$ of GLE showed growth performance parameters not statistically significant with respect to the control ($p > 0.05$). No mortality occurred during the trial. The survival rate was 100% in all groups.

**Table 3.** Growth performance parameters.

|  | CT | GLE1 | GLE2 | GLE3 |
|---|---|---|---|---|
| IW (mg) | 124.1 ± 9.01 [a] | 115.5 ± 7.09 [a] | 130.2 ± 10.07 [a] | 110.3 ± 11.06 [a] |
| FW (mg) | 366.3 ± 5.7 [b] | 348.6 ± 9.2 [b] | 394.2 ± 3.9 [a] | 339.3 ± 10.3 [b] |
| WG (mg) | 242.3 ± 16.3 [b] | 265.6 ± 19.7 [b] | 293.6 ± 9.9 [a] | 229.7 ± 15.6 [b] |
| SGR (% d$^{-1}$) | 1.57 ± 0.13 [b] | 1.68 ± 0.16 [b] | 1.88 ± 0.08 [a] | 1.49 ± 0.19 [b] |

CT = control group; GLE1 = fish fed with 0.5 g kg$^{-1}$ of *Vitis vinifera* leaf extract; GLE2 = fish fed with 1 g kg$^{-1}$ of *Vitis vinifera* extract; GLE3 = fish fed with 2 g kg$^{-1}$ of *Vitis vinifera* extract. Data assigned with different letters in a row denote a significant difference ($p < 0.05$). IW = initial weight; FW = final weight; WG = weight gain; SGR = specific growth rate.

*3.2. Innate Immune Parameters*

Table 4 displays the effects of different levels of GLE on immune parameters in zebrafish WBE and skin mucus. Zebrafish-treated groups showed the highest ($p < 0.05$) levels of WBE and skin mucus total protein. Total Ig and lysozyme activity significantly increased in WBE ($p < 0.05$) but not in the skin mucus ($p > 0.05$) with respect to the control group.

**Table 4.** Effects of different levels of GLE on immune parameters in zebrafish WBE and skin mucus.

| WBE | CT | GLE1 | GLE2 | GLE3 |
|---|---|---|---|---|
| Total protein (g dL$^{-1}$) | 0.10 ± 0.01 [c] | 0.45 ± 0.02 [b] | 0.51 ± 0.02 [a] | 0.48 ± 0.05 [ab] |
| Total Ig (g dL$^{-1}$) | 0.06 ± 0.005 [b] | 0.28 ± 0.009 [a] | 0.31 ± 0.010 [a] | 0.31 ± 0.033 [a] |
| Lysozyme (U mL$^{-1}$) | 5.10 ± 0.20 [c] | 7.40 ± 0.45 [b] | 9.30 ± 0.21 [a] | 8.20 ± 0.65 [ab] |
| **Skin mucus** | | | | |
| Total protein (g dL$^{-1}$) | 0.09 ± 0.005 [b] | 0.12 ± 0.015 [a] | 0.10 ± 0.01 [a] | 0.10 ± 0.04 [a] |
| Total Ig (g dL$^{-1}$) | 0.058 ± 0.008 | 0.079 ± 0.008 | 0.066 ± 0.012 | 0.060 ± 0.017 |
| Lysozyme (U mL$^{-1}$) | 11.10 ± 0.43 | 11.76 ± 0.28 | 11.20 ± 0.52 | 10.90 ± 0.40 |

CT = control group; GLE1 = fish fed with 0.5 g kg$^{-1}$ of *Vitis vinifera* leaf extract; GLE2 = fish fed with 1 g kg$^{-1}$ of *Vitis vinifera* extract; GLE3 = fish fed with 2 g kg$^{-1}$ of *Vitis vinifera* extract. Data assigned with different letters in a row denote a significant difference ($p < 0.05$).

*3.3. Antioxidant Enzyme Activity*

Zebrafish fed different levels of GLE showed a significant ($p < 0.05$) increase in the activity of the antioxidant enzymes CAT, SOD, and GPx in the WBE and CAT in the skin mucus compared to the control (Table 5). No significant ($p > 0.05$) difference was recorded in the case of the skin mucus GPx activity between the treated groups and the control group. The MDA content in the WBE was significantly ($p < 0.05$) lower in the treated group than

the control group, while no significant ($p > 0.05$) differences were detected in the case of the skin mucus MDA content between the treated groups and the control group (Table 5).

**Table 5.** Antioxidant enzyme activities and MDA content in zebrafish whole-body extract (WBE) and skin mucus.

| WBE | CT | GLE1 | GLE2 | GLE3 |
|---|---|---|---|---|
| CAT (U mL$^{-1}$) | 9.91 ± 0.74 [c] | 14.44 ± 0.46 [b] | 16.59 ± 0.29 [a] | 15.93 ± 0.87 [ab] |
| SOD (U mL$^{-1}$) | 520.97 ± 2.03 [b] | 725.05 ± 11.67 [a] | 718.29 ± 3.83 [a] | 724.36 ± 9.62 [a] |
| GPx (U mL$^{-1}$) | 42.55 ± 1.02 [b] | 96.64 ± 1.59 [a] | 96.49 ± 1.59 [a] | 95.50 ± 2.36 [a] |
| MDA (μmol/mL) | 4.60 ± 0.30 [b] | 2.49 ± 0.72 [a] | 2.34 ± 0.90 [a] | 2.35 ± 0.95 [a] |
| **Skin mucus** | | | | |
| CAT (U mL$^{-1}$) | 10.09 ± 0.36 [b] | 11.62 ± 0.65 [a] | 11.57 ± 0.41 [a] | 9.77 ± 0.51 [b] |
| SOD (U mL$^{-1}$) | 530.86 ± 8.03 [ab] | 558.16 ± 13.04 [a] | 548.46 ± 27.69 [aa] | 520.36 ± 15.51 [b] |
| GPx (U mL$^{-1}$) | 42.70 ± 2.40 | 42.53 ± 1.12 | 43.39 ± 1.40 | 42.35 ± 2.10 |
| MDA (μmol/mL) | 4.48 ± 0.84 | 4.06 ± 0.36 | 4.43 ± 0.18 | 4.44 ± 0.45 |

CT = control group; GLE1 = fish fed with 0.5 g kg$^{-1}$ of *Vitis vinifera* leaf extract; GLE2 = fish fed with 1 g kg$^{-1}$ of *Vitis vinifera* extract; GLE3 = fish fed with 2 g kg$^{-1}$ of *Vitis vinifera* extract. Different letters in a row denote significant differences ($p < 0.05$). SOD = Superoxide Dismutase, CAT = Catalase, GPx = Glutathione Peroxidase. MDA = malondialdehyde.

### 3.4. Immune and Oxidative Gene Expression

Figure 1 shows that GLE affected the expression of TNFα and TLR-1 in the gastrointestinal tract of zebrafish. In the group added with 2 g kg$^{-1}$ of GLE, TNFα gene expression appeared to be significantly ($p < 0.05$) upregulated compared to the control group, as well as in the groups fed with 0.5 and 1 g kg$^{-1}$ of GLE. Zebrafish fed with 1 or 2 g kg$^{-1}$ of GLE showed a significant increase in TLR-1 gene expression ($p < 0.05$) compared to the group fed with 0.5 g kg$^{-1}$ of GLE and the control group.

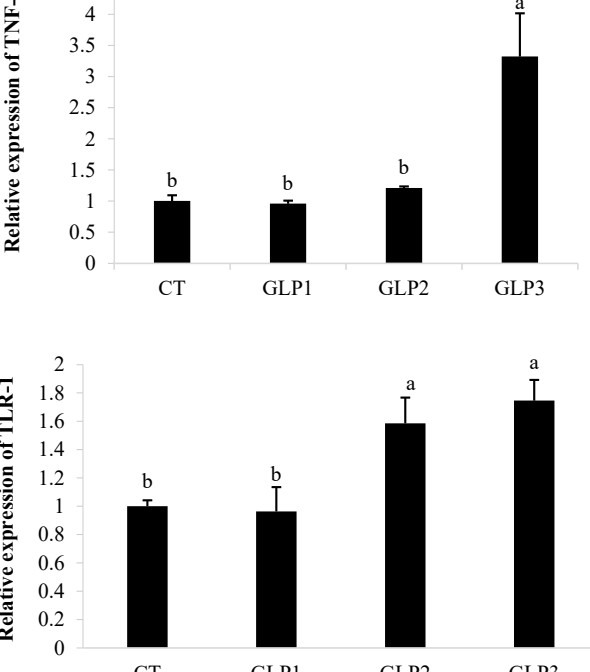

**Figure 1.** Tumor necrosis factor-α (TNF-α) and Toll-like receptor-1 (TLR-1) gene expression in zebrafish gastrointestinal tract. CT = control group; GLE1 = fish fed with 0.5 g kg$^{-1}$ of *Vitis vinifera* leaf extract; GLE2 = fish fed with 1 g kg$^{-1}$ of *Vitis vinifera* leaf extract; GLE3: fish fed with 2 g kg$^{-1}$ of *Vitis vinifera* leaf extract. Bars assigned with different letters denote significant differences ($p < 0.05$).

Zebrafish added with 1 and 2 g kg$^{-1}$ of GLE had significantly ($p < 0.05$) higher CAT and SOD gene expressions in the gastrointestinal of zebrafish with respect to the control and the 0.5 g kg$^{-1}$ GLE-treated groups (Figure 2). Only 1 g kg$^{-1}$ of GLE-added diet significantly increased the expression of the GPx gene ($p < 0.05$), with respect to the control group and the other treated groups (Figure 2).

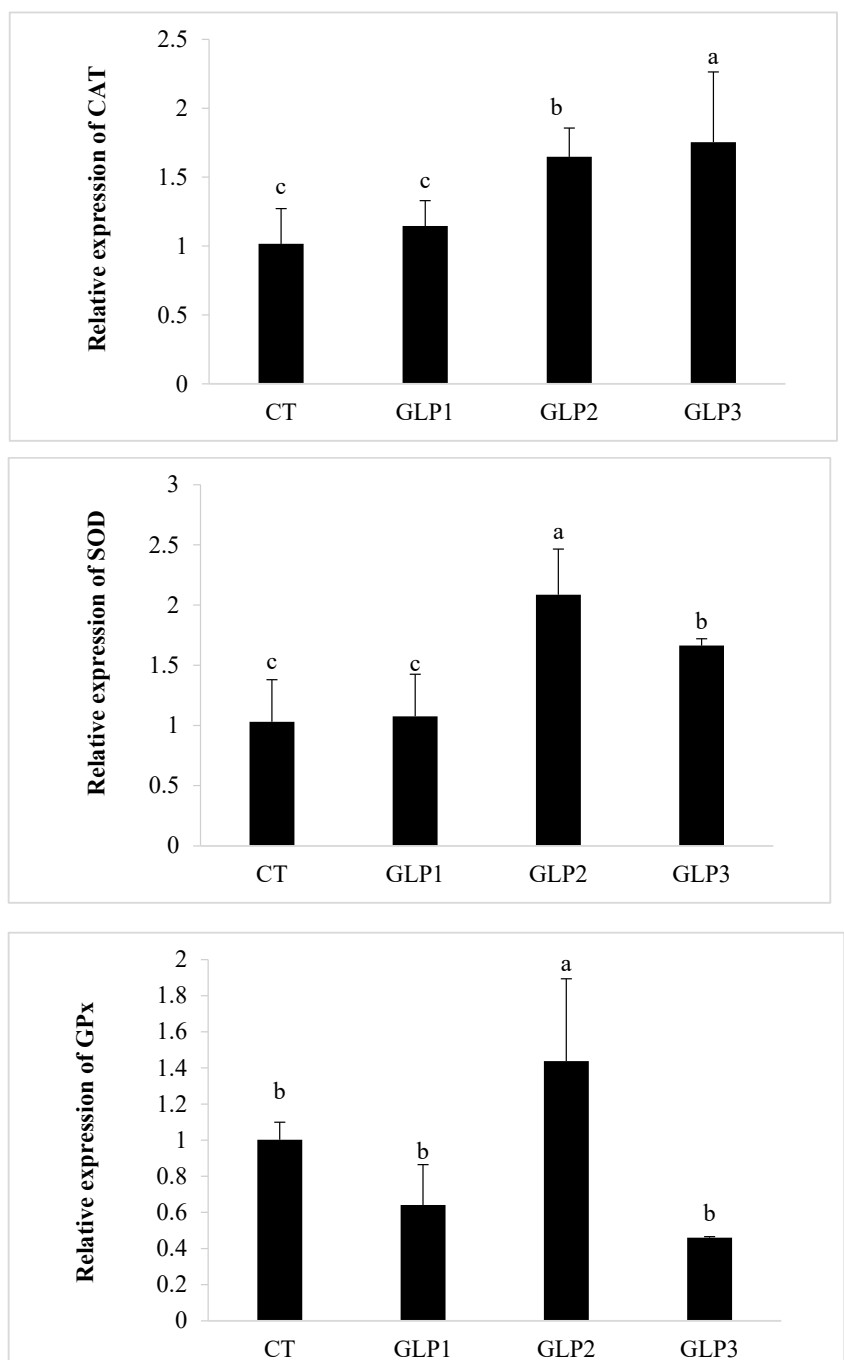

**Figure 2.** Catalase (CAT), Superoxide Dismutase (SOD), and Glutathione Peroxidase (GPx) gene expression in zebrafish gastrointestinal tract. CT = control group; GLE1 = fish fed with 0.5 g kg$^{-1}$ of *Vitis vinifera* leaf extract; GLE2 = fish fed with 1 g kg$^{-1}$ of *Vitis vinifera* leaf extract; GLE3 = fish fed with 2 g kg$^{-1}$ of *Vitis vinifera* leaf extract. Bars assigned with different letters denote significant differences ($p < 0.05$).

## 4. Discussion

In this study, we report the effects of grapevine (*Vitis vinifera* L.) leaf extract on the growth performance, antioxidant status, and immunity of zebrafish (*Danio rerio*).

Natural products such as plant extracts perform many pro-health functions by virtue of the polyphenol content and are widely used in aquaculture [14,37]. However, standardization is still far away and critical issues remain due to the natural variability in the polyphenol content, different extraction techniques, and lack of chemical characterization of polyphenols, which prevent reaching general conclusions about the type and optimal doses of the extracts. In this study, we used a commercially available GLE with a claimed polyphenol content of 5%. Caftaric acid, quercetin, kaempferol, catechins, and rutin are the major phenolic compounds. reported in vine leaves [38–40].

When fish live in suboptimal conditions, they stop growing [41]. The evaluation of the growth performance is therefore crucial to establish the goodness of treatment and is a common welfare marker in aquaculture. The outcome of the present study indicates that zebrafish fed with GLE had improved growth performance. It has been reported in the literature that plant extract inclusion in the diet has positive effects on aquatic species' growth performance. Olive mill wastewater (OMWW) extract, rich in hydroxytyrosol, one of the most powerful natural antioxidants, brought about the growth improvement of the crayfish (*Astacus leptodactylus*) [42]. Chestnut wood extract rich in tannins and OMWW extract significantly increased the growth and feed efficiency in Nile tilapia [16], beluga sturgeon [17], and common carp [18]. On the contrary, in the study by Omnes et al. [20], the diet added with different doses of tannins had a negative impact on the growth and feed conversion ratio in the sea bass. In another study, Sicuro et al. [22,23] found that a diet containing different levels of OMWW extract negatively affected the growth parameters in rainbow trout. Such inconsistencies are likely caused by the different doses of polyphenols present in the extracts employed in the studies, further sustaining that the amount of polyphenols is crucial to the achievement of the expected benefits. The polyphenol doses employed in the present study are indeed lower than those employed by Omnes et al. [20] and Sicuro et al. [21–23], and in agreement with those usually employed as fish feed integrators reported in the literature [37].

Free radicals or ROS (reactive oxygen species) are produced by cells in response to numerous metabolic activities and are normally neutralized by endogenous antioxidants such as glutathione [43]. The overproduction of ROS within cells is a negative but possible event if the cells are exposed to stressful or toxic agents and lead to an increase in oxidative stress [44], an important biomarker in modern aquaculture. The present study reveals that GLE increased the serum and mucus activity of CAT, SOD, and GPx, crucial enzymes responsible for removing excessive free radicals [45]. Such an outcome could be explained by the antioxidant properties able to neutralize ROS and stimulate oxidative stress enzymes of polyphenols [46]. Our findings are in line with the literature indicating that polyphenols administered to farmed fish improve antioxidant defenses [15].

Polyphenols extracted from grapes modulated the antioxidant-relevant gene expression in trout [47]. Previous studies showed that GLE's antioxidant activity is due to the high amount of quercetin [48]. In zebrafish exposed to different levels of quercetin, CAT, SOD, and GPx activity and gene expression increased, reaching the highest value with 1 μg/L of quercetin [49]. In zebrafish exposed to the pro-oxidant triphenyltin, a slight increase in SOD but not in CAT and GPX activity was detected in quercetin-pretreated zebrafish [50]. In silver catfish (*Rhamdia quelen*), the activity of CAT, SOD, and GPX was significantly higher in tissues of fish fed with diets containing quercetin [51].

Parallel to the increase in the antioxidant enzymes, a noticeable decrease in MDA took place. MDS is a chemical compound produced as a consequence of lipid peroxidation and widely employed as a mark of oxidative stress and indicator of peroxidative tissue damage [52]. This is not surprising, since the antioxidant enzymes CAT, SOD, and GPx not only remove excessive ROS but also act on the reduction in lipid peroxidation damage [53]. Indeed, according to our data, quercetin decreased the MDA content in zebrafish [49].

Moreover, tea polyphenols improved the antioxidant enzyme activity and decreased MDA levels in Wuchang bream (*Megalobrama amblycephala*) [54], and apple polyphenols decreased the MDA content while increasing CAT gene expression in grass carp (*Ctenopharyngodon idellus*) [55].

Some of the most appropriate innate immune indicators of fish health status and immune response are lysozyme activity and immunoglobulins levels [56,57]. In the present study, zebrafish-fed diets added with GLE showed increased total proteins, total Ig levels, and lysozyme activity. These results are in agreement with the literature data reporting that polyphenols improve innate immune parameters in Nile tilapia [16], Beluga sturgeon [17], common carp [18], and convict cichlid (*Amatitlamia nigrofasciata*) [19].

Another important class of proteins that play a key role in innate immunity is represented by Toll-like receptors, belonging to the pattern recognition receptors (PRRs), essential proteins capable of recognizing conserved molecules collectively known as microbial/pathogen/danger-associated molecular patterns [58]. PRRs trigger the signaling pathways, including the Nf-Kb pathway, leading to signaling molecule activation of the innate immune response via chemokines, cytokines, antimicrobial peptides, and interferons [59]. In zebrafish (present study), the TLR-1 gene was upregulated by polyphenols, which could partially explain the increase in TNF$\alpha$ as a result of the Nf-Kb pathway activation. This line of reasoning is sustained by Zhang et al. [50], reporting that in zebrafish treated with quercetin-enriched diets, the inflammation decreased by Nf-kB signaling pathway regulation. Altogether, the increase in TNF$\alpha$ and TLR-1 can be regarded as an improvement of the zebrafish's innate immune system to resist pathogen attacks. In the present study, the zebrafish treated with 2 g kg$^{-1}$ of GLE showed a significant increase in the gene expression of TNF$\alpha$—a pro-inflammatory cytokine—as well as TLR-1. Phytochemicals have been found to increase pro-inflammatory cytokines (TNF$\alpha$, IL-1, and IL-8) in fish [14,18]. TNF$\alpha$ in fish is upregulated during the early stages of infection when its intervention is crucial to promote phagocytosis and anti-bactericidal activities [60]. Thus, the transcription of pro-inflammatory cytokines may be considered advantageous to ameliorate fish resistance against pathogens and maintain immunological response. Similarly, a higher relative expression of the TNF$\alpha$ gene was noticed in the Nile tilapia fed with fenugreek (*Trigonella foenum-graecum* L.) [61], and *Leucas aspera*-added diets [62]. Lemon verbena (*Aloysia citrodora*) added 2% upregulated TNF$\alpha$ and IL-8 in rainbow trout, while Apple (*Malus pomila*) upregulated TNF$\alpha$, IL-1, and IL-8 [63]. Dried lemon peel administered at 2.5 and 5% upregulated TNF$\alpha$, IL-1, and IL-8 in the head kidney of *Labeo rohita* [64].

## 5. Conclusions

In conclusion, considering the data obtained, it is thought that adding 0.5–1 g kg of *Vitis vinifera* extract polyphenols to the feeds affects the growth performance of fish, improves the immune system, and fortifies the antioxidant defense system. Investigating the effect of supplements under normal environmental conditions might seem superfluous, as many studies evaluate its effects in suboptimal conditions (stress, temperature, hypoxia, etc.), and/or in the presence of infections. However, it is extremely important to understand the effects of integrators under normal environmental conditions to evaluate the dose and duration of therapy that provides the best protection. In future studies, it will be useful to investigate the effectiveness of *Vitis vinifera* extracts on aquatic organisms exposed to biotic and abiotic stress factors.

**Author Contributions:** Conceptualization, S.H.H. and M.P.; methodology, S.H.H., M.Y. (Morteza Yousefi) and H.V.D.; formal analysis, M.Y. (Morteza Yousefi); investigation, Z.F.; resources, S.H.H., H.V.D., M.Y. (Morteza Yousefi) and M.P.; data curation, M.Y. (Metin Yazici); writing—original draft preparation, E.E.-H.; writing—review and editing, S.H.H. and M.P.; supervision, M.P.; funding acquisition, S.H.H., H.V.D., M.Y. (Morteza Yousefi) and M.P. All authors have read and agreed to the published version of the manuscript.

**Funding:** This research received funding from the University of Sannio (funds FRA number 2022 to Marina Paolucci), Chiang Mai university, and GUASNR.

**Institutional Review Board Statement:** Committee of ethics of the faculty of sciences of the University of Tehran; Approval code 357.

**Informed Consent Statement:** Not applicable.

**Data Availability Statement:** Data available upon request.

**Conflicts of Interest:** The authors declare no conflict of interest.

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
