# Peer review of "The Effects of Grapevine (Vitis vinifera L.) Leaf Extract on Growth Performance, Antioxidant Status, and Immunity of Zebrafish (Danio rerio)"

_fishes, doi:10.3390/fishes8060326_

Round 1
Reviewer 1 Report
MDPI
Manuscript: Fishes-2376692
In this manuscript, the authors reported grapevine leaf extract (GLE) effects on zebrafish growth performance, antioxidant status, and immunity. The authors prepared four diets containing GLE at 0, 0.5, 1, and 2 g kg-1. Fish were fed these diets, respectively for 8 weeks.
The authors proximate composition of the basal diet (Table 1), primers f immune response and antioxidant related genes, TLR-1, TNF--α, GPx, SOD, CAT (Table 2), chromatogram of Vitis vinifera leaf extracts (Fig. 1), growth performance (Table 3), immune parameters of WBE and skin mucus including total protein, total Ig, and lysozyme (Table 4), antioxidant enzyme activity of WBE and skin mucus including CAT, SOD, GPx, MDA (Table 5), relative gene expressions of TNF-α, TLR-1 (Fig. 2), relative expressions of CAT, SOD, GPx (Fig.3). The authors concluded and reported that GLE has the potential to be used as a feed additive in zebrafish to boost growth performance and regulate antioxidant and immune gene expression.
1. Abstract, Line 22: delete “of feed”.
2. Abstract, Line 23: Rewrite the sentence
3. Abstract, Lines 25-27: Rewrite the sentence. Change Lysozyme, Catalase (CAT), Superoxide Dismutase (SOD), Glutathione Peroxidase (GPx) to lysozyme, catalase (CAT), superoxide dismutase (SOD), glutathione peroxidase (GPx).
4. Abstract, Lines 27-32: Rewrite the sentences. Change Malondialdehyde ( MDA) activity to malondialdehyde (MDA) activity.
5. Abstract, Lines 32-33, Last sentence: The statement is not correct. Only fish fed GLE at 1 g kg-1 boost growth performance. Fish in the GLE1, GLE2, and GLE3 groups had higher CAT, GPx, and GPx, but had lower MDA in whole-body extract (WBE). Fish in the GLE1 and GLE2 groups had higher CAT and SOD in sin mucus (Table 5). Fish in the GLE3 group had higher relative expressions of TNF-α and TLR-1 (Fig. 2), Fish in the GLE3 group had higher relative expression of CAT, fish in the GLE 2 and GLE3 groups had higher relative expression of SOD, and fish in the GLE2 group had higher but fish in the GLE3 had lower relative expression of GPx.
6. 2.5: Suggest rewrite the sentence. Need to write clearly about the total protein, Ig, SOD, CAT, GPx assays, and cite appropriate references
7. 2.7: Need to rewrite the first sentence, and last sentence, and cite appropriate references.
8. Discussion is lengthy and disjointed.
9. Conclusion: Suggest rewrite the conclusion. What is the statement “by Imperatore et al. (2023)?
10. Table 5: Suggest check and conduct statistical analysis, and mark significance difference among the treatments.
11. Fig. 2 and Fig. 3, X-axis: Report what CT, GLP1 GLP2, GLP are?
12. Fig. 3: Suggest check and conduct statistical analysis, and mark significance difference among the treatments.
13. References: Suggest check and follow the Journal guide, write references.
MDPI
Manuscript: Fishes-2376692
In this manuscript, the authors reported grapevine leaf extract (GLE) effects on zebrafish growth performance, antioxidant status, and immunity. The authors prepared four diets containing GLE at 0, 0.5, 1, and 2 g kg-1. Fish were fed these diets, respectively for 8 weeks.
The authors proximate composition of the basal diet (Table 1), primers f immune response and antioxidant related genes, TLR-1, TNF--α, GPx, SOD, CAT (Table 2), chromatogram of Vitis vinifera leaf extracts (Fig. 1), growth performance (Table 3), immune parameters of WBE and skin mucus including total protein, total Ig, and lysozyme (Table 4), antioxidant enzyme activity of WBE and skin mucus including CAT, SOD, GPx, MDA (Table 5), relative gene expressions of TNF-α, TLR-1 (Fig. 2), relative expressions of CAT, SOD, GPx (Fig.3). The authors concluded and reported that GLE has the potential to be used as a feed additive in zebrafish to boost growth performance and regulate antioxidant and immune gene expression.
1. Abstract, Line 22: delete “of feed”.
2. Abstract, Line 23: Rewrite the sentence
3. Abstract, Lines 25-27: Rewrite the sentence. Change Lysozyme, Catalase (CAT), Superoxide Dismutase (SOD), Glutathione Peroxidase (GPx) to lysozyme, catalase (CAT), superoxide dismutase (SOD), glutathione peroxidase (GPx).
4. Abstract, Lines 27-32: Rewrite the sentences. Change Malondialdehyde ( MDA) activity to malondialdehyde (MDA) activity.
5. Abstract, Lines 32-33, Last sentence: The statement is not correct. Only fish fed GLE at 1 g kg-1 boost growth performance. Fish in the GLE1, GLE2, and GLE3 groups had higher CAT, GPx, and GPx, but had lower MDA in whole-body extract (WBE). Fish in the GLE1 and GLE2 groups had higher CAT and SOD in sin mucus (Table 5). Fish in the GLE3 group had higher relative expressions of TNF-α and TLR-1 (Fig. 2), Fish in the GLE3 group had higher relative expression of CAT, fish in the GLE 2 and GLE3 groups had higher relative expression of SOD, and fish in the GLE2 group had higher but fish in the GLE3 had lower relative expression of GPx.
6. 2.5: Suggest rewrite the sentence. Need to write clearly about the total protein, Ig, SOD, CAT, GPx assays, and cite appropriate references
7. 2.7: Need to rewrite the first sentence, and last sentence, and cite appropriate references.
8. Discussion is lengthy and disjointed.
9. Conclusion: Suggest rewrite the conclusion. What is the statement “by Imperatore et al. (2023)?
10. Table 5: Suggest check and conduct statistical analysis, and mark significance difference among the treatments.
11. Fig. 2 and Fig. 3, X-axis: Report what CT, GLP1 GLP2, GLP are?
12. Fig. 3: Suggest check and conduct statistical analysis, and mark significance difference among the treatments.
13. References: Suggest check and follow the Journal guide, write references.
Author Response
Manuscript: Fishes-2376692 YELLOW in the ms
In this manuscript, the authors reported grapevine leaf extract (GLE) effects on zebrafish growth performance, antioxidant status, and immunity. The authors prepared four diets containing GLE at 0, 0.5, 1, and 2 g kg-1. Fish were fed these diets, respectively for 8 weeks.
The authors proximate composition of the basal diet (Table 1), primers f immune response and antioxidant related genes, TLR-1, TNF--α, GPx, SOD, CAT (Table 2), chromatogram of Vitis vinifera leaf extracts (Fig. 1), growth performance (Table 3), immune parameters of WBE and skin mucus including total protein, total Ig, and lysozyme (Table 4), antioxidant enzyme activity of WBE and skin mucus including CAT, SOD, GPx, MDA (Table 5), relative gene expressions of TNF-α, TLR-1 (Fig. 2), relative expressions of CAT, SOD, GPx (Fig.3). The authors concluded and reported that GLE has the potential to be used as a feed additive in zebrafish to boost growth performance and regulate antioxidant and immune gene expression.
- Abstract, Line 22: delete “of feed”.
It was deleted
- Abstract, Line 23: Rewrite the sentence
It was revised
- Abstract, Lines 25-27: Rewrite the sentence. Change Lysozyme, Catalase (CAT), Superoxide Dismutase (SOD), Glutathione Peroxidase (GPx) to lysozyme, catalase (CAT), superoxide dismutase (SOD), glutathione peroxidase (GPx).
It was revised
- Abstract, Lines 27-32: Rewrite the sentences. Change Malondialdehyde ( MDA) activity to malondialdehyde (MDA) activity.
It was revised
- 5.Abstract, Lines 32-33, Last sentence: The statement is not correct. Only fish fed GLE at 1 g kg-1boost growth performance. Fish in the GLE1, GLE2, and GLE3 groups had higher CAT, GPx, and GPx, but had lower MDA in whole-body extract (WBE). Fish in the GLE1 and GLE2 groups had higher CAT and SOD in sin mucus (Table 5). Fish in the GLE3 group had higher relative expressions of TNF-α and TLR-1 (Fig. 2), Fish in the GLE3 group had higher relative expression of CAT, fish in the GLE 2 and GLE3 groups had higher relative expression of SOD, and fish in the GLE2 group had higher but fish in the GLE3 had lower relative expression of GPx.
It was revised
- 2.5: Suggest rewrite the sentence. Need to write clearly about the total protein, Ig, SOD, CAT, GPx assays, and cite appropriate references
It was revised as per your suggestion. For antioxidant enzymes we used commercial kit as mentioned.
- 2.7: Need to rewrite the first sentence, and last sentence, and cite appropriate references.
It was revised
- Discussion is lengthy and disjointed.
We did our best to shorten and revise the discussion, hope this will be acceptable.
- Conclusion: Suggest rewrite the conclusion. What is the statement “by Imperatore et al. (2023)?
The conclusion has been totally revised as per your suggestion
- Table 5: Suggest check and conduct statistical analysis, and mark significance difference among the treatments.
We have done statistical analysis and the data with significant difference marked by letter
- Fig. 2 and Fig. 3, X-axis: Report what CT, GLP1 GLP2, GLP are?
It has been declared in figures caption
- 12. 3: Suggest check and conduct statistical analysis, and mark significance difference among the treatments.
We have done statistical analysis and the data with significant difference marked by letter
- References: Suggest check and follow the Journal guide, write references.
Many thanks for your considerations and efforts for evaluation of the manuscript
Reviewer 2 Report
It was a pleasure to review the manuscript provided by Hoseinifar et al. on the effects of Grapevine leaf extract in zebrafish. In my opinion, this manuscript provides enough novel information to be published in the journal; however, the following major corrections should be addressed in the revised version:
The title of manuscript should be changed to “The effects of grapevine (Vitis vinifera L.) leaf extract on growth performance, antioxidant status, and immunity of zebrafish (Danio rerio)”.
Line 22. Please change “oxidative enzyme activity” to “oxidative enzymatic activities”.
Lines 22-23. Please change “antioxidant gene expression” to “expression of antioxidant genes”.
Line 23. Please change “360” to “A total of 360”.
Line 28. Please change “decrease the Malondialdehyde” to “decrease in the Malondialdehyde”.
Lines 40-50. The first paragraph of introduction should be removed.
Line 58. Please change “their effects protective” to “their protective effects”.
Lines 68-69. Please rewrite “we can identify”.
Line 73. “in vitro” should be italic.
Line 79. Please rewrite “is still at the dawn”.
Lines 84, 88, 90, 133, etc. The expression of all units should be identical throughout the manuscript according to the format requested by the journal.
Line 85. This sentence should be rewritten.
Line 87. “and” should be added between common carp and convict cichlid.
Line 95. Please modify “was done preventively”.
Line 108-110. Please rewrite the sentence.
Line 113. Please rewrite “innate immunity, and immune”.
Line 116. The ethic number assigned to the study should be added.
Line 120. Please elaborate on the method used for supplementation of basal diet with Grapevine leaf extract.
Line 122. Please add a relevant reference. Also, please add the size of commercial diet used.
Ine 122. Please change “with” to “for”.
Lines 124-125. Please add here the product name used.
Line 128. Please change “purchased in” to “purchased from”.
Line 133. Please specify the correct full name abbreviated “Vitis vinifera leaf spray-dried extract (GLE)”.
Lines 133, 134 and 135. Please change to “Kg-1 of Vitis vinifera” to “Kg-1 of feed Vitis vinifera”. This should be corrected throughout the manuscript. Also please change “added” to “supplemented”.
Line 137. Please change “At most care” to “Utmost care”?!
Line 149. Please change “Compound” to “The compound”.
Lines 154-156. Please rewrite.
Ines 163-164. Please specify the number of mucus samples taken (n=9).
Line 172. Please change “from each treatment nine fish” to “nine fish from each treatment”.
Line 174. Please rewrite.
Line 181. Please change “cDNA of samples was prepared” to “cDNA samples were prepared”.
Line 184. The method used for normalization of gene expression data should be added with a reference.
Lines 186-187. Please rewrite “Sequence and melting temperature (Tm) primers of selected mucosal immune response 186 and antioxidant related genes expression in Zebrafish”.
In Table 2, please provide the correct accession number for TNF‐α. Also, please remove “.1” from the end of accession numbers.
Line 190. The normality test used should be added.
Table 3. The authors should clarify why the initial weight of zebrafish is remarkably different between the groups. There is 20 mg difference between group 4 with group 3 (15-20 % of the body weight), which could also affect the final weight of fish.
Line 223. Please correct “in the serum”.
Lines 227-229. This lines should be removed.
Line 224. Please rewrite “Immune and oxidative gene expression”.
Line 246. A comma should be added after “GLE”.
Line 250. Please change “with the group” to “to the group”.
Line 151. Please remove “diets”.
Line 266. Should be “zebrafish gastrointestinal tract”.
Line 271-272. Please rewrite.
Line 184. Please replace “growth performances” to “growth performance” throughout the manuscript.
Line 292, 294, 299, 346. Please add the name of the authors mentioned.
Line 306. Oxidative stress is not an important biomarker?!
Lines 309-310. Please rewrite.
Line 333. Please change “show” to “showed”.
Please change “” to “”.
Please change “” to “”.
Line 366-374. Please be precise and provide a scientific conclusion according to the findings obtained in your study. These sentences should be removed or moved to the section introduction.
The English writing of the manuscript should be improved.
Author Response
GREEN in the ms
It was a pleasure to review the manuscript provided by Hoseinifar et al. on the effects of Grapevine leaf extract in zebrafish. In my opinion, this manuscript provides enough novel information to be published in the journal; however, the following major corrections should be addressed in the revised version:
The title of manuscript should be changed to “The effects of grapevine (Vitis vinifera L.) leaf extract on growth performance, antioxidant status, and immunity of zebrafish (Danio rerio)”.
Done
Line 22. Please change “oxidative enzyme activity” to “oxidative enzymatic activities”.
Done
Lines 22-23. Please change “antioxidant gene expression” to “expression of antioxidant genes”.
Done
Line 23. Please change “360” to “A total of 360”.
Done
Line 28. Please change “decrease the Malondialdehyde” to “decrease in the Malondialdehyde”.
Done
Lines 40-50. The first paragraph of introduction should be removed.
Done
Line 58. Please change “their effects protective” to “their protective effects”.
Done
Lines 68-69. Please rewrite “we can identify”.
Done
Line 73. “in vitro” should be italic.
- We were told that according to the journal policy, in vitro is not in italics
Line 79. Please rewrite “is still at the dawn”.
Done
Lines 84, 88, 90, 133, etc. The expression of all units should be identical throughout the manuscript according to the format requested by the journal.
Checked and corrected
Line 85. This sentence should be rewritten.
Line 87. “and” should be added between common carp and convict cichlid.
The sentence was corrected and shortened
Line 95. Please modify “was done preventively”.
We erased this part of the sentence
Line 108-110. Please rewrite the sentence.
We rewrote the all paragraph
Line 113. Please rewrite “innate immunity, and immune”.
We rewrote and simplified the paragraph
Line 116. The ethic number assigned to the study should be added.
Many thanks for your comments, however, in our university we don’t have ethic commity and ethic number for researchers, to consider ethic we followed the standard ethic framework released by University of Tehran. This has been clarified in ms.
Line 120. Please elaborate on the method used for supplementation of basal diet with Grapevine leaf extract.
It has been considered
Line 122. Please add a relevant reference. Also, please add the size of commercial diet used.
It has been considered
Ine 122. Please change “with” to “for”.
done
Lines 124-125. Please add here the product name used
done.
Line 128. Please change “purchased in” to “purchased from”.
done
Line 133. Please specify the correct full name abbreviated “Vitis vinifera leaf spray-dried extract (GLE)”.
Specified the first time it was mentioned (line 122)
Lines 133, 134 and 135. Please change to “Kg-1 of Vitis vinifera” to “Kg-1 of feed Vitis vinifera”. This should be corrected throughout the manuscript.
Many thanks for your suggestion; however, given different reviewers suggested different changes in this regards, it was not possible to consider.
Also please change “added” to “supplemented”.
done
Line 137. Please change “At most care” to “Utmost care”?!
The phrase was reformulated
Line 149. Please change “Compound” to “The compound”.
done
Lines 154-156. Please rewrite.
It has been considered
Ines 163-164. Please specify the number of mucus samples taken (n=9).
done
Line 172. Please change “from each treatment nine fish” to “nine fish from each treatment”.
done
Line 174. Please rewrite.
done
Line 181. Please change “cDNA of samples was prepared” to “cDNA samples were prepared”.
done
Line 184. The method used for normalization of gene expression data should be added with a reference.
It has been provided
Lines 186-187. Please rewrite “Sequence and melting temperature (Tm) primers of selected mucosal immune response 186 and antioxidant related genes expression in Zebrafish”.
It has been revised
In Table 2, please provide the correct accession number for TNF‐α. Also, please remove “.1” from the end of accession numbers.
It has been considered
Line 190. The normality test used should be added.
It has been added
Table 3. The authors should clarify why the initial weight of zebrafish is remarkably different between the groups. There is 20 mg difference between group 4 with group 3 (15-20 % of the body weight), which could also affect the final weight of fish.
Thanks for your comments, it was not possible for such a little fish to keep weight identical and difference is inevitable. The main important point is that there was no significant difference between initial weight of the treatments. We don’t think that such non-significant difference could affect the final weight.
Line 223. Please correct “in the serum”.
done
Lines 227-229. This lines should be removed.
done
Line 224. Please rewrite “Immune and oxidative gene expression”.
We couldn’t find such term in line 224
Line 246. A comma should be added after “GLE”.
done
Line 250. Please change “with the group” to “to the group”.
done
Line 151. Please remove “diets”.
done
Line 266. Should be “zebrafish gastrointestinal tract”.
done
Line 271-272. Please rewrite.
done
Line 184. Please replace “growth performances” to “growth performance” throughout the manuscript.
done
Line 292, 294, 299, 346. Please add the name of the authors mentioned.
Done
Line 306. Oxidative stress is not an important biomarker?!
Yes it is indeed
Lines 309-310. Please rewrite.
done
Line 333. Please change “show” to “showed”.
done
Line 366-374. Please be precise and provide a scientific conclusion according to the findings obtained in your study. These sentences should be removed or moved to the section introduction.
We have totally re-written the conclusion
Reviewer 3 Report
This piece of research evaluates the effect of supplementing diets for zebrafish with an extract obtained from Vitis vinifera leafs. The subject is of interest owing to this kind of extracts are rich in polyphenolic extracts. Results are also interesting, but there are several concerns that auuthors should consider for improving the quality of the manuscript. The following aspects sholud be addressed by authors in the revised version.
In the manuscript title modify: “Growth performance”
Line 23 and 24: Use kg instead of Kg
Lines 27, 29 “activities”
Keywords by alphabetical order.
Line 77. Sort in italic letters: Saccharomyces cerevisiae
Line 90. Check unit in “gr/kg”
Line 131. Table 2.
Line 133. The bioactive extract is liquid and the supplementation is detailed in g/kg, Should the supplementation be expresssed in mL kg?. How 2 g of the bioactive extract were applied to 1 kg of basal feed? Auhtors must provide more information about the procedure for incoorporating the bioactive extracts into aquafeeds.
Line 140. Provide the list of ingredients of the basal diet, and pellet size. Authors should also provide information about how the bioactive extract was included in the basal feed.
Line 152. Knowing the concentration of bioactive in the leaf extract is important but the effective concentration of polyphenols in the experimental aquafeeds is quite relevant too. Are these results available?
Line 155. Specify concentration of clove oil used for this purpose
Line 162. Specify if were used nine fish per tank or per treatment. Specify the number of different whole-body extracts prepared per dietary treatment.
Lines 161-170. Description of methods is quite scarce, and more details should be provided. Volume of fish extract used in each assay, time for enzyme reaction, substrates, etc.
Line 197. This figure confirmed the presence of the bioactive compounds but the concentration of each one is the important information from a practical point of view. That information should be provided instead of a typical chromatogram evidencing the presence of these polyphenols. Even, the effective concentration in experimental aquafeeds is also relevant.
Line 215. The description of dietary treatment should be detailed in the table footnote instead of the heading of the table.
Table headings, description of tables should be shortened. For instance, Table3 information related to description of treatment and growth parameters should be moved to table footnote, and Table 4 heading includes a description of results. This information should be omitted from the table heading.
Table 4. Include a table footnote detailing “mean values with different letter in a row denotes significant difference (P<0.05). For Lysozyme activity sort units as for the rest of parameters (U mL-1). For instance, g kg-1 mg L-1.
Table 5, The same comments as before. Activiy of enzymes should be expressed in U /mL. MDA is not an enzyme, it is a product resulting from the oxidation of lipid, so modify the table heading.
Table 5. SOD activity in skin mucus in control group is 530.86 ± 8.03 b and in GLE2 is 548.46 ± 27.69 a. It is hard to undertand how these values are different owing to the reported SD in the GLE2 treatment. Clarify this point, please.
Figures 2 and 3. In the Y-axis decimal position should be indicated with “dot” in the ciphers. Sort Vitis vinifera in italic letters, check in the whole document.
Lines 251-255. Figure 3 is not cited in the text.
Results from Fig 2 and 3 should be presented in a single Table.
Line 363 Authors should avoid to include bibliographical references in the conclusion paragraph.
Overall, the document is well written and easy for reading.
Author Response
Reviewer 3 pale blue in the ms
Your work is well contextualized with the current attempt to use industry/agriculture by-products to improve animal farming and increase production yield.
Regarding the introduction I think it is well framed with the problem/objective. In line 58: "their effects protective" - protective effects
done
line 86: "tannins from chestnut shell" is repeated.
The all period was rephrased
Regarding the objective: it should be more articulated with the conclusion. You do not directly answer the question of "evaluate the role of polyphenols..."
The all period was rephrased
In line 113: innate immunity and immune are repeated.
The all period was rephrased
In results:
GROWTH- In SGR no differences were found between diets, despite differences in FW. How do you justify this? All the results show a tendency to increase with GLE except GLE3. Could be a dosage effect?
Thanks for your comment, regarding the growth performance parameters, as we mentioned in text and marked the statistical analysis difference by letter, all of the parameters in GLE2 treatment were significantly higher than control and other treatments. No significant difference was observed among GL3, GLE1 and control.
Gene expression - You see the same pattern with GLE3 as above. You do not address this.
In case of gene expression results, almost in all treatments GLE2 and 3 should better results. Given that the best growth results were observed in GLE2, seems that we can suggest this level
Why were the analysis done only in intestine? include the info for that in the text, since liver could also be targeted for antioxidant status.
Thanks for your suggestion, we agree with you that we could consider other tissues as well, however, the main limitations for us were funding, and we had to extract RNA and synthesize cDNA with just one tissue. This was done in ours and others studies as well. We will consider your valuable point in future studies.
In line 332- Ig and Immunoglobulins repeated.
corrected
Discussion is well written, with reference to similar works.
As stated above, conclusion of this work should be clearer and separated from the future perspectives.
Many thanks for your suggestion, the conclusion section has been revised based on yours and other reviewers’ suggestion
Reviewer 4 Report
Your work is well contextualized with the current attempt to use industry/agriculture by-products to improve animal farming and increase production yield.
Regarding the introduction I think it is well framed with the problem/objective. In line 58: "their effects protective" - protective effects
line 86: "tannins from chestnut shell" is repeated.
Regarding the objective: it should be more articulated with the conclusion. You do not directly answer the question of "evaluate the role of polyphenols..."
In line 113: innate immunity and immune are repeated.
In results:
GROWTH- In SGR no differences were found between diets, despite differences in FW. How do you justify this? All the results show a tendency to increase with GLE except GLE3. Could be a dosage effect?
Gene expression - You see the same pattern with GLE3 as above. You do not address this.
Why were the analysis done only in intestine? include the info for that in the text, since liver could also be targeted for antioxidant status.
In line 332- Ig and Immunoglobulins repeated.
Discussion is well written, with reference to similar works.
As stated above, conclusion of this work should be clearer and separated from the future perspectives.
Author Response
This piece of research evaluates the effect of supplementing diets for zebrafish with an extract obtained from Vitis vinifera leafs. The subject is of interest owing to this kind of extracts are rich in polyphenolic extracts. Results are also interesting, but there are several concerns that auuthors should consider for improving the quality of the manuscript. The following aspects sholud be addressed by authors in the revised version.
In the manuscript title modify: “Growth performance”
We couldn’t get the points raised by the respected reviewer; we used this term as it was used in many of the same papers.
Line 23 and 24: Use kg instead of Kg
It has been considered
Keywords by alphabetical order.
It has been considered
Line 77. Sort in italic letters: Saccharomyces cerevisiae
It has been considered
Line 90. Check unit in “gr/kg”
It has been considered
Line 131. Table 2.
That was out mistake, it has been corrected
Line 133. The bioactive extract is liquid and the supplementation is detailed in g/kg, Should the supplementation be expresssed in mL kg?. How 2 g of the bioactive extract were applied to 1 kg of basal feed? Authors must provide more information about the procedure for incoorporating the bioactive extracts into aquafeeds.
Many thanks for your comment We received the product in powder form, indeed, the product sent to market as powder to ease administration. Regarding preparation of experimental diet, we have mentioned in text.
Line 140. Provide the list of ingredients of the basal diet, and pellet size. Authors should also provide information about how the bioactive extract was included in the basal feed.
Many thanks for your suggestion. We used as commercial diet as basal diet, so the ingredients are not public, and we just mentioned the proximate composition. Also, we revised the text as per your suggestion and make it clear how we prepared experimental diet.
Line 152. Knowing the concentration of bioactive in the leaf extract is important but the effective concentration of polyphenols in the experimental aquafeeds is quite relevant too. Are these results available?
Many thanks for your valuable comments, we do agree with you, unfortunately, we have considered this due to limitation in fud.
Line 155. Specify concentration of clove oil used for this purpose
It has been considered
Line 162. Specify if were used nine fish per tank or per treatment. Specify the number of different whole-body extracts prepared per dietary treatment.
It has been considered
Lines 161-170. Description of methods is quite scarce, and more details should be provided. Volume of fish extract used in each assay, time for enzyme reaction, substrates, etc.
We did our best to revise this part.
Line 197. This figure confirmed the presence of the bioactive compounds but the concentration of each one is the important information from a practical point of view. That information should be provided instead of a typical chromatogram evidencing the presence of these polyphenols. Even, the effective concentration in experimental aquafeeds is also relevant.
Many thanks for your valuable comments, we do agree with you that that providing the quantitative amounts of the bioactive compounds would be nice in the present study. However, due to limitations in fund, we couldn’t perform such test. We will consider your valuable point in future studies.
Line 215. The description of dietary treatment should be detailed in the table footnote instead of the heading of the table.
It has been considered
Table headings, description of tables should be shortened. For instance, Table3 information related to description of treatment and growth parameters should be moved to table footnote, and Table 4 heading includes a description of results. This information should be omitted from the table heading.
It has been considered
Table 4. Include a table footnote detailing “mean values with different letter in a row denotes significant difference (P<0.05). For Lysozyme activity sort units as for the rest of parameters (U mL-1). For instance, g kg-1 mg L-1.
It has been considered
Table 5, The same comments as before. Activiy of enzymes should be expressed in U /mL. MDA is not an enzyme, it is a product resulting from the oxidation of lipid, so modify the table heading.
It has been revised
Table 5. SOD activity in skin mucus in control group is 530.86 ± 8.03 b and in GLE2 is 548.46 ± 27.69 a. It is hard to understand how these values are different owing to the reported SD in the GLE2 treatment. Clarify this point, please.
Many thanks for your precious view, we have checked the statistical analysis again and found that there is a typus in the table, we have corrected the mistake in table
Figures 2 and 3. In the Y-axis decimal position should be indicated with “dot” in the ciphers. Sort Vitis vinifera in italic letters, check in the whole document.
It has been considered
Lines 251-255. Figure 3 is not cited in the text.
It has been considered
Results from Fig 2 and 3 should be presented in a single Table.
It has been considered
Line 363 Authors should avoid to include bibliographical references in the conclusion paragraph.
It has been considered
Round 2
Reviewer 1 Report
MDPI
Manuscript: Fishes-2376692
In this manuscript, the authors reported grapevine leaf extract (GLE) effects on zebrafish growth performance, antioxidant status, and immunity. The authors prepared four diets containing GLE at 0, 0.5, 1, and 2 g kg-1. Fish were fed these diets, respectively for 8 weeks.
The authors proximate composition of the basal diet (Table 1), primers f immune response and antioxidant related genes, TLR-1, TNF--α, GPx, SOD, CAT (Table 2), chromatogram of Vitis vinifera leaf extracts (Fig. 1), growth performance (Table 3), immune parameters of WBE and skin mucus including total protein, total Ig, and lysozyme (Table 4), antioxidant enzyme activity of WBE and skin mucus including CAT, SOD, GPx, MDA (Table 5), relative gene expressions of TNF-α, TLR-1 (Fig. 2), relative expressions of CAT, SOD, GPx (Fig.3). The authors concluded and reported that GLE has the potential to be used as a feed additive in zebrafish to boost growth performance and regulate antioxidant and immune gene expression.
1. Abstract, Line 22: delete “of feed”.
2. Abstract, Line 23: Rewrite the sentence
3. Abstract, Lines 25-27: Rewrite the sentence. Change Lysozyme, Catalase (CAT), Superoxide Dismutase (SOD), Glutathione Peroxidase (GPx) to lysozyme, catalase (CAT), superoxide dismutase (SOD), glutathione peroxidase (GPx).
4. Abstract, Lines 27-32: Rewrite the sentences. Change Malondialdehyde ( MDA) activity to malondialdehyde (MDA) activity.
5. Abstract, Lines 32-33, Last sentence: The statement is not correct. Only fish fed GLE at 1 g kg-1 boost growth performance. Fish in the GLE1, GLE2, and GLE3 groups had higher CAT, GPx, and GPx, but had lower MDA in whole-body extract (WBE). Fish in the GLE1 and GLE2 groups had higher CAT and SOD in sin mucus (Table 5). Fish in the GLE3 group had higher relative expressions of TNF-α and TLR-1 (Fig. 2), Fish in the GLE3 group had higher relative expression of CAT, fish in the GLE 2 and GLE3 groups had higher relative expression of SOD, and fish in the GLE2 group had higher but fish in the GLE3 had lower relative expression of GPx.
6. 2.5: Suggest rewrite the sentence. Need to write clearly about the total protein, Ig, SOD, CAT, GPx assays, and cite appropriate references
7. 2.7: Need to rewrite the first sentence, and last sentence, and cite appropriate references.
8. Discussion is lengthy and disjointed.
9. Conclusion: Suggest rewrite the conclusion. What is the statement “by Imperatore et al. (2023)?
10. Table 5: Suggest check and conduct statistical analysis, and mark significance difference among the treatments.
11. Fig. 2 and Fig. 3, X-axis: Report what CT, GLP1 GLP2, GLP are?
12. Fig. 3: Suggest check and conduct statistical analysis, and mark significance difference among the treatments.
13. References: Suggest check and follow the Journal guide, write references.
Author Response
Many thanks for your comments and suggestion. All the points raised here by the respected reviewer were considered and replied to in the first round. The point-by-point responses is :
In this manuscript, the authors reported grapevine leaf extract (GLE) effects on zebrafish growth performance, antioxidant status, and immunity. The authors prepared four diets containing GLE at 0, 0.5, 1, and 2 g kg-1. Fish were fed these diets, respectively for 8 weeks.
The authors proximate composition of the basal diet (Table 1), primers f immune response and antioxidant related genes, TLR-1, TNF--α, GPx, SOD, CAT (Table 2), chromatogram of Vitis vinifera leaf extracts (Fig. 1), growth performance (Table 3), immune parameters of WBE and skin mucus including total protein, total Ig, and lysozyme (Table 4), antioxidant enzyme activity of WBE and skin mucus including CAT, SOD, GPx, MDA (Table 5), relative gene expressions of TNF-α, TLR-1 (Fig. 2), relative expressions of CAT, SOD, GPx (Fig.3). The authors concluded and reported that GLE has the potential to be used as a feed additive in zebrafish to boost growth performance and regulate antioxidant and immune gene expression.
- Abstract, Line 22: delete “of feed”.
It was deleted
- Abstract, Line 23: Rewrite the sentence
It was revised
- Abstract, Lines 25-27: Rewrite the sentence. Change Lysozyme, Catalase (CAT), Superoxide Dismutase (SOD), Glutathione Peroxidase (GPx) to lysozyme, catalase (CAT), superoxide dismutase (SOD), glutathione peroxidase (GPx).
It was revised
- Abstract, Lines 27-32: Rewrite the sentences. Change Malondialdehyde ( MDA) activity to malondialdehyde (MDA) activity.
It was revised
- 5.Abstract, Lines 32-33, Last sentence: The statement is not correct. Only fish fed GLE at 1 g kg-1boost growth performance. Fish in the GLE1, GLE2, and GLE3 groups had higher CAT, GPx, and GPx, but had lower MDA in whole-body extract (WBE). Fish in the GLE1 and GLE2 groups had higher CAT and SOD in sin mucus (Table 5). Fish in the GLE3 group had higher relative expressions of TNF-α and TLR-1 (Fig. 2), Fish in the GLE3 group had higher relative expression of CAT, fish in the GLE 2 and GLE3 groups had higher relative expression of SOD, and fish in the GLE2 group had higher but fish in the GLE3 had lower relative expression of GPx.
It was revised
- 2.5: Suggest rewrite the sentence. Need to write clearly about the total protein, Ig, SOD, CAT, GPx assays, and cite appropriate references
It was revised as per your suggestion. For antioxidant enzymes we used commercial kit as mentioned.
- 2.7: Need to rewrite the first sentence, and last sentence, and cite appropriate references.
It was revised
- Discussion is lengthy and disjointed.
We did our best to shorten and revise the discussion, hope this will be acceptable.
- Conclusion: Suggest rewrite the conclusion. What is the statement “by Imperatore et al. (2023)?
The conclusion has been totally revised as per your suggestion
- Table 5: Suggest check and conduct statistical analysis, and mark significance difference among the treatments.
We have done statistical analysis and the data with significant differences marked by letters
- Fig. 2 and Fig. 3, X-axis: Report what CT, GLP1 GLP2, GLP are?
It has been declared in figures' caption
- 12. 3: Suggest check and conduct statistical analysis, and mark significance difference among the treatments.
We have done statistical analysis and the data with significant differences marked by letters
- References: Suggest checking and following the Journal guide, and write references.
Many thanks for your consideration and efforts for evaluation of the manuscript
Reviewer 2 Report
The authors have provided the required revisions, and the manuscript is now suggested to be published in the journal.
One minor edit:
Line 275. "growth performance" should be removed from the end of this sentence.
Author Response
Many thanks for your comments which helped us to improve the ms. We have considered the minor point.
Reviewer 3 Report
Authors have paid attention to the reviewer comments but there are two point that remain unadressed:
-My main concern with Fig 1 is this figure just detail the qualitative profile of poliphenols in the extract so my question is; is this need to include that figure in the manuscriopt? Just details the compounds detected in the extract when describing the additive would be enough. I consider this figure must be omitted from the manuscript since it is not revelant. Really relevant would be the quantitification of each polyphenol in the additive or knowing the real concentration of each one (or total polyhenols) in the experimental aquafeeds (ppm of mg kg feed). This can help to compare with other studies focussing on mg of active compound kg diets instead g of additive pef kg diet.
-Figures 2 and 3. In the Y-axis decimal position must be detailed with dots in the ciphres.
Author Response
Authors have paid attention to the reviewer comments but there are two point that remain unadressed:
Many thanks for your valuable comments
-My main concern with Fig 1 is this figure just detail the qualitative profile of poliphenols in the extract so my question is; is this need to include that figure in the manuscriopt? Just details the compounds detected in the extract when describing the additive would be enough. I consider this figure must be omitted from the manuscript since it is not revelant. Really relevant would be the quantitification of each polyphenol in the additive or knowing the real concentration of each one (or total polyhenols) in the experimental aquafeeds (ppm of mg kg feed). This can help to compare with other studies focussing on mg of active compound kg diets instead g of additive pef kg diet.e have
We have removdfig1 as per your suggestion
-Figures 2 and 3. In the Y-axis decimal position must be detailed with dots in the ciphres.
We couldn't understand the point raised by the respected reviewers, We already used dots for decimals,